# Development and Validation of a Prognostic Model for Overall Survival in Patients with Primary Pelvis and Spine Osteosarcoma: A Population-Based Study and External Validation

**DOI:** 10.3390/jcm12072521

**Published:** 2023-03-27

**Authors:** Da Wang, Fanrong Liu, Binbin Li, Jinhui Xu, Haiyi Gong, Minglei Yang, Wei Wan, Jian Jiao, Yujie Liu, Jianru Xiao

**Affiliations:** 1Department of Orthopedic Oncology, Shanghai Changzheng Hospital, Navy Military Medical University, Shanghai 200003, China; 2Department of Orthopedics, The First Affiliated Hospital of Wenzhou Medical University, Wenzhou 325015, China; 3Department of Pathology, Shanghai Changzheng Hospital, Navy Military Medical University, Shanghai 200003, China

**Keywords:** osteosarcoma, spine, pelvis, overall survival, nomogram

## Abstract

Background: Primary pelvis and spine osteosarcoma (PSOS) is a specific type of osteosarcoma that is difficult to treat and has a poor prognosis. In recent years, the research on osteosarcoma has been increasing, but there have been few studies on PSOS; in particular, there have been a lack of analyses with a large sample size. This study aimed to construct and validate a model to predict the overall survival (OS) of PSOS patients, as currently there are no tools available for assessing their prognosis. Methods: Data including demographic information, clinical characteristics, and follow-up information on patients with PSOS were collected from the Surveillance, Epidemiology, and End Results (SEER) database, as well as from the Spine Tumor Center of Changzheng Hospital. Variable selection was achieved through a backward procedure based on the Akaike Information Criterion (AIC). Prognostic factors were identified by univariate and multivariate Cox analysis. A nomogram was further constructed for the estimation of 1-, 3-, and 5-year OS. Calibration plots, the concordance index (C-index), and the receiver operating characteristic (ROC) were used to evaluate the prediction model. Results: In total, 83 PSOS patients and 90 PSOS patients were separately collected from the SEER database and Changzheng Hospital. In the SEER cohort, liver metastasis, lung metastasis, and chemotherapy were recognized as independent prognostic factors for OS (*p* < 0.05) and were incorporated to construct the initial nomogram. However, the initial nomogram showed poor predictive accuracy in internal and external validation. Then, we shifted our focus to the Changzheng data. Lung metastasis involving segments, Eastern Cooperative Oncology Group (ECOG) performance score, alkaline phosphatase (ALP) level, and en bloc resection were ultimately identified as independent prognostic factors for OS (*p* < 0.05) and were further incorporated to construct the current nomogram, of which the bias-corrected C-index was 0.834 (0.824–0.856). The areas under the ROC curves (AUCs) of the current nomogram regarding 1-, 3-, and 5-year OS probabilities were 0.93, 0.96, and 0.92, respectively. Conclusion: We have developed a predictive model with satisfactory performance and clinical practicability, enabling effective prediction of the OS of PSOS patients and aiding clinicians in decision-making.

## 1. Introduction

Primary pelvis and spine osteosarcoma (PSOS) is a rare malignancy constituting less than 10% of all osteosarcomas [1,2,3,4,5]. Osteosarcomas are the most common primary malignant bone tumors, accounting for 20% of all cases of primary malignant bone tumors in the world, and they occur primarily in the metaphysis of the long tubular bones but are rare in the spine and pelvis [6]. Patients with PSOS are reported to have a worse prognosis than those with cancer in other primary sites [7,8,9]. Skeletal-related events secondary to PSOS, such as intractable pain, pathological fractures, and neurological deficits, have great impacts on patients’ health.

Nowadays, surgical resection combined with adjuvant chemotherapy is the generally accepted treatment for osteosarcoma patients. However, treating PSOS is a difficult task due to its complex anatomical structure and high rate of local recurrence [10,11,12]. When treating PSOS patients, clinicians must consider both their life expectancy and functional status. Unfortunately, the currently existing scoring systems have different limitations and are not appropriate for the estimation of prognosis in PSOS patients.

The Surveillance, Epidemiology, and End Results (SEER) program database is a comprehensive population-level cancer database comprising data from 18 registries covering 28% of the US population. It offers advantages, such as multi-center data and a large patient pool, that improve its statistical power. The Spine Tumor Center of Changzheng Hospital is one of the largest facilities in China dedicated to the treatment of spinal tumors.

A nomogram is a graphical model that combines key factors to generate a personalized estimation of the probability of any event. [13,14]. Furthermore, nomograms are easy to use and have been proven to be a reliable tool for predicting the prognosis of different cancers, such as breast cancer, lung cancer, and liver cancer [15,16,17]. When we conducted our research, there were few studies on PSOS and no predictive models for PSOS patients.

Given the rarity, poor prognosis, and lack of a predictive model, it is necessary to develop a model for prognosis estimation in PSOS patients. Detailed clinical characteristics, such as patient status, tumor stage, and treatment strategy, should be taken into account. Our study aimed to tackle this issue. This article follows the STROBE reporting checklist.

## 2. Materials and Methods

### 2.1. Study Design and Participants

Based on the SEER database (http://seer.cancer.gov/, accession number: 12794-Nov2021, accessed on 11 May 2022), we extracted the data of patients pathologically diagnosed with osteosarcoma from 2010 to 2018, and these were named as the SEER cohort through the SEER*STAT software (version 8.3.9). In addition, under the approval of the ethics committee of Changzheng Hospital, its database was retrospectively examined to gather data on patients pathologically diagnosed with osteosarcoma between 2010 and 2018, and these were named as the Changzheng cohort.

Inclusion criteria were as follows: (I) confirmed diagnosis of osteosarcoma as a primary tumor (International Classification of Diseases for Oncology ICD-O-3 histology code: 9180-9187); (II) diagnosis between 1 January 2010 and 31 December 2018; (III) acquired diagnosis while alive; and (IV) spine or pelvis as the primary site. In the SEER database, the codes for the primary site were C412 (vertebral column) and C414 (pelvic bones, sacrum, coccyx, and associated joints).

Patients were excluded based on the following criteria: (I) other pathological types: in the Changzheng cohort, pathologic diagnoses were made by two pathologists independently; hematoxylin and eosin (HE) stainings were considered as the gold standard, with immunohistochemistry (IHC) results also being evaluated; cases without IHC results to support the diagnosis were excluded; (II) spine or pelvis was not the primary site: image data including X-ray plain film, computed tomography (CT) scan, magnetic resonance imaging (MRI), or PET-CT were collected and demonstrated; primary sites in extremities or spine metastasis were excluded; (III) incomplete data collection: patients with inadequate relevant data, including unknown tumor size, unconfirmed stage, survival month of 0, and lost to follow-up.

### 2.2. Data Collection

This study followed the Helsinki Declaration of 1975, and written informed consent was waived due to the retrospective study design. Features in both cohorts included demographic characteristics, pathological diagnosis, clinical characteristics, treatment methods, follow-up period, and survival outcomes. Compared with the SEER database, the Changzheng database contained more clinical characteristics, including family history, medical records, laboratory results imaging results, functional status, and surgical approach. Age was stratified into three groups according to the epidemiology of osteosarcoma. In accordance with the 8th edition of the American Joint Committee on Cancer (AJCC) staging system, patients were divided into different clinical stages [18]. The cutoff point for tumor size was decided according to the previously published literature, and the cutoff point for the first intervention was decided by the optimal cutoff value [19]. As important bone tumor marker indexes, alkaline phosphatase (ALP) and lactate dehydrogenase (LDH) were applied in this study, with the definition of normal and abnormal levels described in previous papers [20,21,22]. Intensity of symptoms was measured according to the Visual Analogue Scale (VAS). Neurological function was classified using the Frankel score at admission. The performance status of the patients was assessed according to their loss of weight and Eastern Cooperative Oncology Group (ECOG) performance score. Surgery, radiotherapy (RTx), and/or chemotherapy (CTx) were administered in accordance with interdisciplinary guidance. En bloc resection was selected as a potential prognostic factor. En bloc resection appears to particularly improve recurrence-free and overall survival for aggressive primary tumors. Patients were typically informed of these methods and made an autonomous choice regarding their therapy in the informed consent discussion.

This study utilized a standard follow-up procedure. All patients were followed up monthly in the first year and every three months thereafter through outpatient visits and telephone interviews. Enhanced CT or MRI was conducted on return visits. PET-CT was conducted yearly to detect metastases. The follow-up was from diagnosis to the point at which an event of interest occurred or the deadline. The deadline of follow-up was 31 December 2018. The endpoint in the study was overall survival (OS).

### 2.3. Statistical Analysis

Quantitative data were characterized by median values and interquartile ranges (IQRs), and qualitative data were characterized by counts with percentages. Continuous variables were compared by the Wilcoxon Mann–Whitney test, while categorical variables were compared by the Pearson χ^2^ test or Fisher’s exact test. The X-tile software (version 3.6.1) was utilized to define the optimal cutoff values of continuous variables. Variables were analyzed to exclude multicollinearity in SPSS by using the variance inflation factor (VIF). Univariable and multivariable Cox regression analysis followed by a stepwise regression were performed in order to determine independent predictors by calculating the hazard ratio (HR) with a 95% confidence interval (CI). These predictors were then incorporated into a multivariate Cox regression model. A nomogram was generated to interpret the Cox regression model and predict 1-, 3-, and 5-year OS probabilities. The discriminative ability was evaluated by calculating C-indices and the area under curve (AUC) of the receiver operating characteristic (ROC) curves. C-indices were internally validated through bootstrapping with 1000 resamples, and calibration plots were generated for accuracy assessment through bootstrapping with 1000 resamples. Statistical analyses were performed with the X-tile software (Yale University, New Haven, CT, USA), SPSS25.0 (SPSS Inc., Chicago, IL, USA), and R software (version 4.0.1). R packages including survival, survminer, rms, ggplot2, timeROC, etc., were used. Two-sided *p* < 0.05 was regarded statistically significant.

## 3. Results

### 3.1. Patient Baseline Characteristics

A total of 127 patients diagnosed with PSOS were identified in the SEER database between 2010 and 2018 based on the inclusion criteria, of whom 44 patients were excluded based on the exclusion criteria. Eventually, 83 patients were enrolled in the SEER cohort as the training cohort. From the Spine Tumor Center of Changzheng Hospital, we obtained the Changzheng cohort, which totaled 90 patients, who were assigned as the validation cohort. Flowcharts depicting the step-by-step screening of the two cohorts are shown in Figure 1.

Table 1 provides the clinicopathological characteristics of the two cohorts. The training cohort had a median follow-up of 14 months (7–33 months), and that of the validation cohort was 37.5 months (20–61.25 months). The median age at the diagnosis of PSOS was 42 years (22–67 years) for the training cohort and 30 years (18.75–47.25 years) for the validation cohort. The median tumor size was 100 mm (mm) (65–130 mm) for the training cohort and 58.5 mm (44.25–79.25 mm) for the validation cohort.

In the training cohort, 39 patients (47%) were female and 44 patients (53%) were male. Moreover, 31 patients (37%) were married and 52 patients (63%) had another marital status. One patient (1%) was diagnosed with Grade I, which meant well-differentiated tumor cells; one patient (1%) was Grade II, or moderately differentiated; and sixty-four patients (77%) were Grade III and IV, which meant poorly differentiated. One patient (1%) had liver metastasis and nineteen patients (23%) had lung metastasis. Forty-two (51%) patients underwent surgery, twenty-eight (34%) underwent radiotherapy, and sixty-seven (81%) underwent chemotherapy. Until the deadline of follow-up, 47 patients (57%) had died, which was attributed to any cause.

In the validation cohort, 35 patients (39%) were female and 55 patients (61%) were male. Moreover, 46 patients (51%) were married and 44 patients (49%) had another marital status. Nine patients (10%) were diagnosed with Grade I, fifty-one patients (57%) were Grade II, and thirty patients (33%) were Grade III and IV. One patient (1%) had liver metastasis and twenty-four patients (27%) had lung metastasis. Eighty-four (93%) had surgery, twenty-one (23%) had radiotherapy, and forty-one (46%) had chemotherapy. Until the deadline of follow-up, 90 patients (100%) had died.

Apart from the same variables, the Changzheng cohort contained numerous additional clinical characteristics, which are individually listed in Table 2. In terms of blood tests at admission, the median ALP level was 175.5 U/L (118.75–211.5 U/L) and the median LDH level was 231 U/L (169.75–297.5 U/L). Regarding the nature of lesions, 24 patients (27%) were osteolytic, 43 patients (48%) were osteoblastic, and 23 patients (25%) were mixed. In terms of medical history, seven patients (8%) had diabetes, thirteen patients (14%) had hypertension, thirteen patients (14%) had hyperlipidemia, sixteen patients (18%) had a smoking history, and six patients (7%) had a family history. Regarding patients’ status at admission, thirty-nine patients (43%) had a weight loss of more than 5 kg; forty-two patients (47%) were classified between A and C according to the Frankel Score; and forty-three patients’ (48%) ECOG performance scores were three or four. Meanwhile, fifty-six patients (62%) experienced severe pain, with scores from seven to ten on the VAS. The median time from symptom onset to the first intervention was four months (2–6.5 months). Of these, sixty-one patients’ (68%) tumors were found only on one part of the spine or pelvis, and fourteen patients (16%) received en bloc resection.

### 3.2. Statistical Analysis of Prognostic Factors in SEER Cohort

The univariate analysis results showed age (*p* = 0.036), liver metastases (*p* = 0.004), lung metastases (*p* < 0.0001), surgery (*p* = 0.001), and chemotherapy (*p* = 0.001) as potential clinical determinants of OS in the training cohort. The above factors were submitted to the multivariate Cox regression model, which revealed that liver metastasis (HR = 17.201, 95%CI = 1.654–178.909, *p* = 0.017), lung metastasis (HR = 3.047, 95%CI = 1.400–6.630, *p* = 0.005), and chemotherapy (HR = 0.296, 95%CI = 0.138–0.633, *p* = 0.002) were independently associated with OS (Table 3).

### 3.3. Development and Validation of the Initial Nomogram

A nomogram was accordingly developed, integrating the three identified independent variables to predict 1-, 3-, and 5-year OS probability (Figure 2). The score for each significant variable is provided in Table 4. The initial nomogram’s performance was internally and externally validated through assessing the discriminative ability and calibration of the model. Bootstrap resampling of the training cohort yielded a bias-corrected C-index of 0.688 (95% CI = 0.657–0.727). The AUCs of the time-dependent ROC curves are shown in Figure 3. The ROC curves showed that the nomogram had poor predictive accuracy regarding 1-, 3-, and 5-year OS rates (AUCs of 0.76, 0.79, and 0.74, respectively). In the validation cohort, the bias-corrected C-index was 0.684 (95% CI = 0.657–0.713). Figure 4 displays the internal and external calibration plots. The calibration plots revealed the initial nomogram’s poor accuracy for predicting 1-, 3-, and 5-year survival rates. The quantity and quality of the data could have affected the accuracy of the model. To provide a more precise estimation, we switched to the data from Changzheng Hospital.

### 3.4. Statistical Analysis of Prognostic Factors in Changzheng Cohort

By selecting features with no multicollinearity (VIF < 5) from the 90 patients in the cohort, we were able to identify 30 candidate variables associated with OS. These candidate variables were further selected using a backward stepwise method with the smallest AIC value. Once selected, 16 potential predictive factors were then involved in the univariable and multivariable Cox regression analysis. Lastly, a multivariate Cox regression model incorporating significant prognostic factors, including lung metastasis (HR = 3.673, 95% CI = 1.525–8.846, *p* = 0.004), involving segments (HR = 3.742, 95% CI = 1.802–7.770, *p* < 0.0001), ALP (HR = 1.799, 95% CI = 1.043–3.105, *p* = 0.035), ECOG (HR = 2.573, 95% CI = 1.373–4.822, *p* = 0.003), and en bloc resection (HR = 0.400, 95% CI = 0.177–0.904, *p* = 0.028), was established (Table 5).

### 3.5. Development and Validation of Changzheng Nomogram

The five finally identified independent variables were included to develop the novel survival estimation nomogram, which was also called the Changzheng nomogram (Figure 5). Table 6 presents the score for each significant variable.

Internal validations of the Changzheng nomogram were performed. Bootstrap resampling yielded an excellent bias-corrected C-index of 0.834 (95% CI = 0.824–0.856), indicating the nomogram’s excellent discriminative ability. The time-dependent ROC curves showed excellent predictive accuracy regarding the 1-, 3-, and 5-year OS rates (AUCs of 0.93, 0.96, and 0.92, respectively) for this nomogram (Figure 6). Bootstrap resampling internally validated the Changzheng nomogram’s appreciable reliability for predicting 1-, 3-, and 5-year survival rates (Figure 7).

## 4. Discussion

PSOS is a rare and aggressive type of osteosarcoma. Vertebral osteosarcomas account for only 1–4% of all osteosarcomas, and pelvic osteosarcomas account for only 7–9% of all osteosarcomas [1,2,3,4,5]. Due to its rarity, there have only been a few published series reporting the oncologic outcomes and surgical therapies for PSOS [7,8,9]. These studies reported very poor survival data, which were much worse than those for extremity osteosarcoma [7,8,9]. In PSOS, the complex anatomical structure and bulky primary lesions make it challenging to resect with an adequate margin and result in a poor response to chemotherapy [10,11,12]. Therefore, the treatment of PSOS patients is challenging and requires caution, making life expectancy prediction critical in clinicians’ decision-making.

Unfortunately, the existing scoring systems have different limitations and are not suitable for the prognostic estimation of patients with PSOS [23,24,25,26,27]. The American Joint Committee on Cancer (AJCC) TNM staging system for osteosarcoma may be confusing because different primary sites have different definitions of TNM categories [18,23]. The Enneking surgical staging system is primarily used in the extremities instead of the spinal column [24]. The Weinstein–Boriani–Biagini (WBB) surgical staging system is widely used for spinal tumors to select the corrected modalities of operation, but it is not suitable for prognostic estimation [25]. The modified Tokuhashi and Tomita scoring systems are used to generally assess patients with spine metastasis rather than primary bone tumors [26,27]. No nomogram model exists for PSOS patients, although nomograms have been confirmed to be an effective tool for the prediction of prognosis in diverse tumors [13,14]. Thus, we aimed to develop a nomogram model for the prediction of the long-term survival of these patients. Clinicians can directly estimate the OS of PSOS patients based on this predictive model.

Given the multi-center and representative data, we first attempted to utilize the SEER database as the training cohort for the construction of the nomogram, and the Changzheng database was used as the validation cohort for external validation. Three clinical features, namely liver metastasis, lung metastasis, and chemotherapy, were incorporated into the initial nomogram. Our initial study found that the diagnosis of liver or lung metastasis was related to a worse survival prognosis for PSOS patients. Similar results were reported in early studies [9]. Our study confirmed that metastasis was significantly related to a shorter OS [28,29]. Given the poor prognosis, a comprehensive diagnostic strategy plays an important role in the screening of primary metastatic diseases for early detection and access to treatment. Our initial study also revealed that chemotherapy was associated with better outcomes for patients with PSOS. This finding was in accordance with recent recommendations [30]. Although osteosarcoma is treated by multimodal approaches, including surgery, chemotherapy, and radiotherapy, chemotherapy plays an irreplaceable role in the treatment of PSOS [31].

However, the initial nomogram showed a poor predictive accuracy in internal and external validation. We surmised that the inaccuracy was caused by the quantity and quality of the data, which affected the model. For example, the SEER database lacked complete records for patients who did not receive chemotherapy and radiation (“no/unknown” for chemotherapy, and “no/unknown” for radiation). Furthermore, many clinical variables were not included in the SEER database. In order to provide a more precise estimation, we shifted our focus to the data from Changzheng Hospital.

The current nomogram, also known as the Changzheng nomogram, had an excellent predictive accuracy regarding 1-, 3-, and 5-year OS rates (AUCs of 0.93, 0.96, and 0.92, respectively). In the Changzheng nomogram, the prediction is derived from the following covariates: lung metastasis, involving segments, ALP, ECOG, and en bloc resection.

The diagnosis of lung metastasis is closely related to a worse survival prognosis for patients with PSOS, which is consistent with our initial study. The lung is the most common metastatic site of osteosarcoma. Once patients with PSOS develop lung metastasis, it results in a worse prognosis. Thus, we suggest that routine chest CT has a significant impact on PSOS patients’ hospital admission and long-term surveillance for early detection and access to treatment for lung metastasis.

Hao et al. reported that a higher level of ALP can decrease the OS in patients with osteosarcoma [32]. Our present study also found that PSOS patients with abnormal ALP levels had a worse prognosis than normal ones [21,22]. Although a high serum ALP level is valuable for the diagnosis and prognosis of osteosarcoma in adults, its use in teenagers is problematic because ALP levels are affected by age and gender. With the rapid development of technology, more and more studies have reported novel markers for poor prognosis of osteosarcoma. Nigris et al. established that overexpression of the polycomb transcription factor Yin Yang 1 (YY1) in the primary site of osteosarcoma is associated with a poor prognosis, which may be a novel marker for patients with osteosarcoma [33]. Zhu et al. identified that circular RNAs (circRNAs), hsa_circ_0081001 and circPVT1, were significantly up-regulated in osteosarcoma and were associated with a poor overall survival. [34,35]. We believe that novel markers will be used as part of routine clinical practice in the future, complementing the prognostic model for PSOS patients.

PSOS patients with an ECOG scale score of three or four had a worse prognosis. This may be due to the worse physical conditions in higher-ECOG-scale patients. These patients were unable to tolerate surgery or long-term chemotherapy. Similar findings were reported in spinal metastasis [36].

The number of involved segments in PSOS is associated with patients’ prognoses. Similar findings were reported in spinal metastasis. Aoude et al. performed a retrospective study of 126 patients, in which they analyzed the individual parameter of the revised Tokuhashi score and evaluated its accuracy in determining patient survival; they concluded that the number of vertebral metastases and metastases to major organs were the paramount predictors of actual survival [36]. The involved segments not only reflect the extent of the tumor but also affects the surgical strategy. For instance, in a PSOS patient whose lesion is localized to one vertebra, surgeons are more likely to attempt an aggressive surgical strategy such as a wide resection if possible.

En bloc resection has been reported to improve the prognosis of primary bone tumors of the spine [25,37,38]. Information about the en bloc resection of PSOS is very limited. Schoenfeld et al. reported seven cases that received en bloc resection in Massachusetts General Hospital [31]. In their study, en bloc resection was not a significant factor associated with survival. Schwab et al. reported nine patients treated with en bloc resection, which only implied that there was a trend toward improved survival with en bloc excision [8]. However, the morbidity of en bloc resection in the spine should be taken into account in decision-making. The complications of en bloc resection may affect the patient’s quality of life and even worsen the prognosis. Boriani et al. performed a retrospective study of 134 patients who had undergone en bloc resection in the spine [39]. Forty-seven of the one hundred and thirty-four patients (34.3%) suffered a total of seventy complications. Three patients (2.2%) died from complications. Demura et al. performed a retrospective study of 307 patients who underwent total en bloc spondylectomy in a single center [40]. Major and minor operative complications were observed in 122 (39.7%) and 84 (27.4%) patients, respectively. These complications included bleeding more than 2000 mL in 60 (19.5%) patients, cerebrospinal fluid leakage in 45 (14.7%), respiratory complications in 52 (16.9%), and cardiovascular complications in 11 (3.6%). Any of these complications may be fatal if not managed effectively. There have been no large series demonstrating the effects of en bloc resection on survival in PSOS. Our study revealed that en bloc resection improved the OS of PSOS patients. Given the site-specific anatomic constraints, appropriate resection margins, and surgical complications, the spine or pelvis pose particular challenges with regard to en bloc resection. The decision-making process for en bloc resection should consider not only its high morbidity but also its positive impact on local control and prognosis in PSOS. Our current study supports the impression that en bloc resection should be recommended for PSOS patients when needed. En bloc resections must be conducted by dedicated teams with trained oncological surgeons and anesthesiologists.

To our knowledge, this is the first study to develop a nomogram to predict the OS of patients with PSOS. However, several limitations of our study should be acknowledged. Firstly, this single-center study had unavoidable bias as a result of its retrospective nature. This retrospective study may have suffered from selection bias in terms of patient inclusion. Secondly, despite the internal validation, the limited sample size of the current nomogram means it lacked external validation, as PSOS is rare. Multi-center research with a larger cohort and prospective double-blind randomized clinical trials are needed to address the gaps in this study.

## 5. Conclusions

Lung metastasis involving segments, ALP, ECOG, and en bloc resection were independent prognostic factors for the OS of PSOS patients. Although some limitations existed, the current nomogram according to these factors presented a remarkable discriminative ability and prediction accuracy to individually predict the survival probability at certain time points for PSOS patients, which could aid in the optimization of clinical decision making.

## Figures and Tables

**Figure 1 jcm-12-02521-f001:**
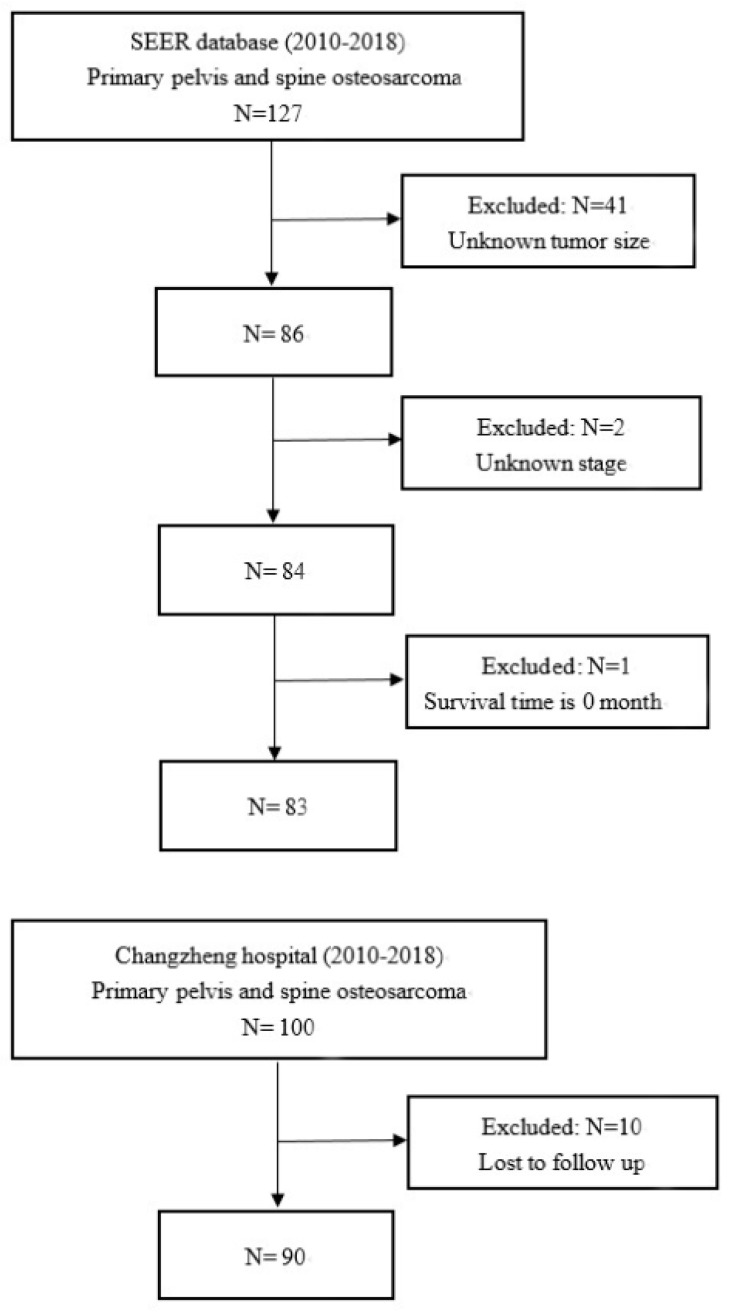
Flowchart showing step–by–step screening of eligible patients.

**Figure 2 jcm-12-02521-f002:**
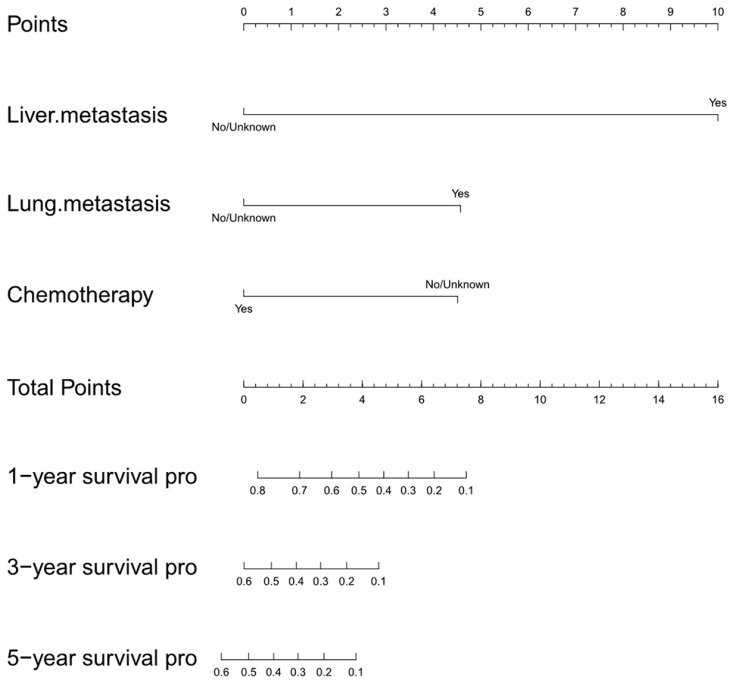
The initial nomogram predicting 1-, 3-, and 5-year OS probability of PSOS patients. Points (0–10) assigned to each clinical variable sum to indicate OS probability at 1, 3, and 5 years. Pro, probability; OS, overall survival.

**Figure 3 jcm-12-02521-f003:**
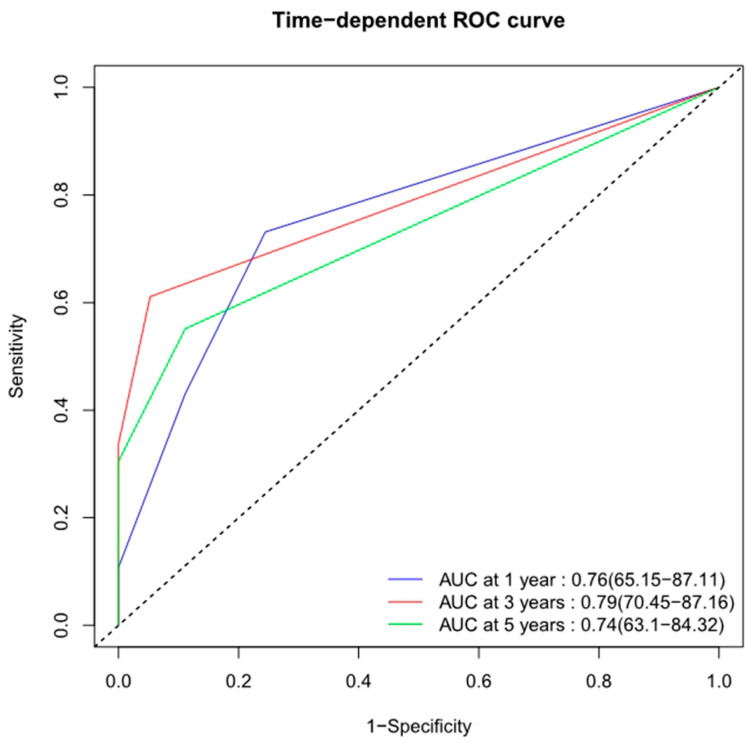
Time-dependent ROC curves of the initial nomogram predicting 1-, 3- and 5-year overall survival with corresponding AUC values. ROC, receiver operating characteristic; AUC, area under curve.

**Figure 4 jcm-12-02521-f004:**
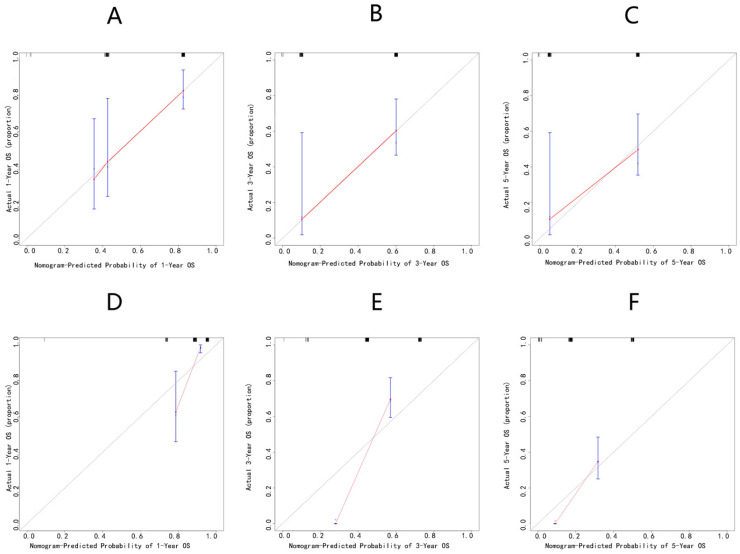
Calibration plots of the initial nomogram in the training and validation cohorts. Overall survival at 1 year (**A**,**D**), 3 years (**B**,**E**), 5 years (**C**,**F**). Data are from the training cohort (**A**–**C**) and from the validation cohort (**D**–**F**). The initial nomogram-predicted probabilities were stratified into equally sized subgroups. For each subgroup, the average nomogram-predicted probability (x-axis) was plotted against the Kaplan–Meier probability observed in the same subgroup (y-axis). The 95% Cis of the Kaplan–Meier estimates are indicated with vertical lines. The continuous line indicates the reference line, where an ideal nomogram would lie.

**Figure 5 jcm-12-02521-f005:**
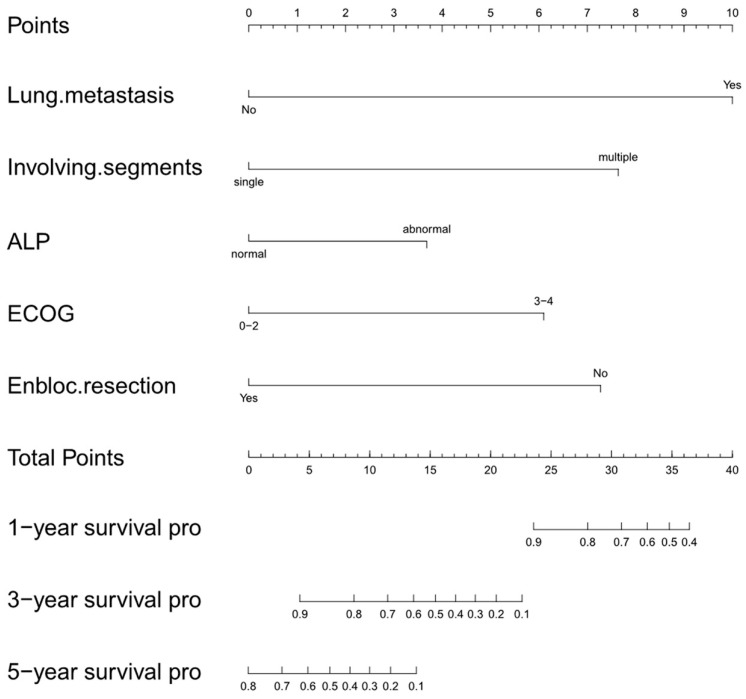
Changzheng nomogram for overall survival prediction of patients with primary pelvis and spine osteosarcoma. The current nomogram calculates 1-, 3-, and 5-year survival probability based on the patient’s lung metastasis, involved segments, ALP, ECOG, and en bloc resection as first events. Points (0–10) assigned to each clinical variable sum to indicate OS probability at 1, 3, and 5 years. Pro, probability; OS, overall survival; ALP, alkaline phosphatase; ECOG, Eastern Cooperative Oncology Group.

**Figure 6 jcm-12-02521-f006:**
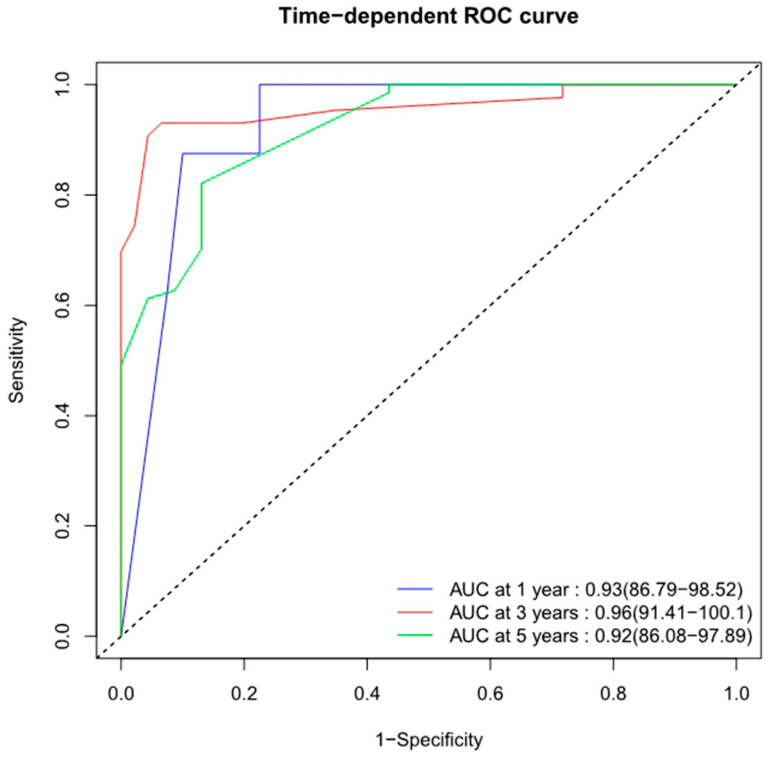
Time-dependent ROC curves of Changzheng nomogram predicting 1-, 3-, and 5-year overall survival with corresponding AUC values. ROC, receiver operating characteristic; AUC, area under curve.

**Figure 7 jcm-12-02521-f007:**
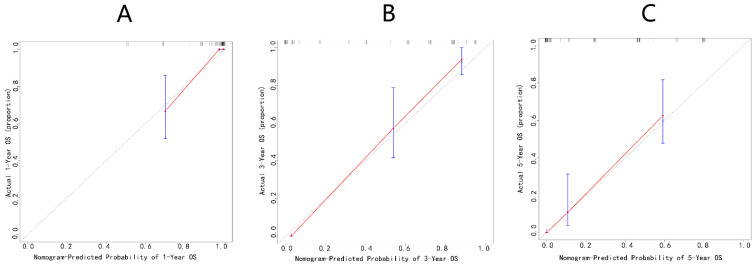
Internal and calibration plots of Changzheng nomogram through bootstrapping with 1000 resamples predicting 1-year (**A**), 3-year (**B**), and 5-year (**C**) overall survival.

**Table 1 jcm-12-02521-t001:** Baseline characteristics of the training cohort and validation cohorts.

Characteristic	Total n = 173 No. (%)	Training Cohort n = 83 No. (%)	Validation Cohort n = 90 No. (%)	*p* Value
Age (yrs, median, IQR)	32 (19–55.5)	42 (22–67)	30 (18.75–47.25)	0.003 *
Sex				0.282
female	74 (43)	39 (47)	35 (39)	
male	99 (57)	44 (53)	55 (61)	
Marital status				0.069
married	77 (45)	31 (37)	46 (51)	
other	96 (55)	52 (63)	44 (49)	
Race recode				<0.0001 *
black	9 (5)	9 (11)	0 (0)	
white	67 (39)	67 (81)	0 (0)	
other	97 (56)	7 (8)	90 (100)	
Laterality				0.225
left	64 (37)	29 (35)	35 (39)	
right	68 (39)	29 (35)	39 (43)	
paired	1 (1)	1 (1)	0 (0)	
not a paired	40 (23)	24 (29)	16 (18)	
Grade				<0.0001 *
unknown	17 (10)	17 (21)	0 (0)	
I	10 (6)	1 (1)	9 (10)	
II	52 (30)	1 (1)	51 (57)	
III and IV	94 (54)	64 (77)	30 (33)	
T				<0.0001 *
Tx	2 (1)	2 (2)	0 (0)	
T1	60 (35)	29 (35)	31 (34)	
T2	52 (30)	46 (55)	6 (7)	
T3	12 (7)	2 (2)	10 (11)	
T4	47 (27)	4 (5)	43 (48)	
N				0.002 *
Nx	6 (3)	6 (7)	0 (0)	
N0	147 (85)	73 (88)	74 (82)	
N1	20 (12)	4 (5)	16 (18)	
Tumor size (mm, median, IQR)	70 (50.25–106.5)	100 (65–130)	58.5 (44.25–79.25)	<0.0001 *
Liver metastasis				0.954
no	171 (99)	82 (99)	89 (99)	
yes	2 (1)	1 (1)	1 (1)	
Lung metastasis				0.566
no	130 (75)	64 (77)	66 (73)	
yes	43 (25)	19 (23)	24 (27)	
Stage				0.134
localized	45 (26)	16 (19)	29 (32)	
regional	81 (47)	41 (49)	40 (44)	
distant	47 (27)	26 (32)	21 (23)	
Surgery				<0.0001 *
no	47 (27)	41 (49)	6 (7)	
yes	126 (73)	42 (51)	84 (93)	
Radiotherapy				0.129
no/unknown	124 (72)	55 (66)	69 (77)	
yes	49 (28)	28 (34)	21 (23)	
Chemotherapy				<0.0001 *
no/unknown	65 (38)	16 (19)	49 (54)	
yes	108 (62)	67 (81)	41 (46)	
Vital status				<0.0001 *
alive	36 (21)	36 (43)	0 (0)	
dead	137 (79)	47 (57)	90 (100)	
Follow up (m, median, IQR)	22 (11–52)	14 (7–33)	37.5 (20–61.25)	<0.0001 *

yrs, years old; IQR, interquartile range; mm, millimeters; m, months. *p* values in Wilcoxon Mann–Whitney test or Fisher’s exact test, depending on whether the variable is continuous or categorical, to test the difference between training cohort and validation cohort. * represents *p* < 0.05.

**Table 2 jcm-12-02521-t002:** Baseline characteristics of the 90 PSOS patients in Changzheng Hospital.

Characteristic	Number of Patients (%)	Characteristic	Number of Patients (%)
Demographics		ALP (u/L, median, IQR)	175.5 (118.75–211.5)
Age (yrs, median, IQR)	30 (18.75–47.25)	LDH (u/L, median, IQR)	231 (169.75–297.5)
Sex		Diabetes	
female	35 (39)	no	83 (92)
male	55 (61)	yes	7 (8)
Marital status		Hypertension	
married	46 (51)	no	77 (86)
other	44 (49)	yes	13 (14)
Clinical features		Hyperlipidemia	
Laterality		no	77 (86)
left	35 (39)	yes	13 (14)
right	39 (43)	Smoking history	
paired	0 (0)	no	74 (82)
not paired	16 (18)	yes	16 (18)
Grade		Family history	
I	9 (10)	no	84 (93)
II	51 (57)	yes	6 (7)
III and IV	30 (33)	Patient status	
T		Loss of weight	
T1	31 (34)	<5 kg	51 (57)
T2	6 (7)	≥5 kg	39 (43)
T3	10 (11)	Frankel	
T4	43 (48)	A–C	42 (47)
N		D–E	48 (53)
N0	74 (82)	ECOG	
N1	16 (18)	0–2	47 (52)
Liver metastasis		3–4	43 (48)
no	89 (99)	VAS	
yes	1 (1)	0–6	34 (38)
Lung metastasis		7–10	56 (62)
no	66 (73)	Treatment	
yes	24 (27)	First intervention (m, median, IQR)	4 (2–6.5)
Brain metastasis		Radiotherapy	
no	86 (96)	no	69 (77)
yes	4 (4)	yes	21 (23)
Stage		Chemotherapy	
localized	29 (32)	no	49 (54)
regional	40 (44)	yes	41 (46)
distant	21 (23)	Surgery	
Medical information		no	6 (7)
Nature of lesions		yes	84 (93)
osteolytic	24 (27)	En bloc resection	
osteoblastic	43 (48)	no	76 (84)
mixed	23 (25)	yes	14 (16)
Involving segments		Vital status	
single	61 (68)	alive	0 (0)
multiple	29 (32)	dead	90 (100)
Tumor size (mm, median, IQR)	58.5 (44.25–79.25)	Follow up (m, median, IQR)	37.5 (20–61.25)

PSOS, primary pelvis and spine osteosarcoma; yrs, years old; IQR, interquartile range; mm, millimeters; m, months; ALP, alkaline phosphatase; LDH, lactate dehydrogenase; kg, kilograms; ECOG, Eastern Cooperative Oncology Group; VAS, Visual Analogue Scale.

**Table 3 jcm-12-02521-t003:** Univariate and multivariate analysis of overall survival in SEER cohort (N = 83).

Prognostic	Univariate Analysis	Multivariate Analysis	
Factors	*p* Value	Hazard Ratio (95% CI)	*p* Value
Age (yrs)	0.036 *		
0–24		Reference	
25–59		1.129 (0.540–2.358)	0.748
≥60		1.499 (0.682–3.295)	0.314
Race recode	0.931		
black			
white			
other			
Sex	0.12		
female			
male			
Marital status	0.478		
married			
other			
Laterality	0.533		
left			
right			
paired			
not a paired			
Grade	0.962		
unknown			
I			
II			
III and IV			
T	0.913		
Tx			
T1			
T2			
T3			
T4			
N	0.262		
Nx			
N0			
N1			
Liver metastasis	0.004 *		
no		Reference	
yes		17.201 (1.654–178.909)	0.017 *
Lung metastasis	<0.0001 *		
no		Reference	
yes		3.047 (1.400–6.630)	0.005 *
Stage	0.082		
localized			
regional			
distant			
Tumor size	0.64		
≤80 mm			
>80 mm			
Surgery	0.001 *		
no		Reference	
yes		0.560 (0.282–1.113)	0.098
Radiotherapy	0.548		
no/unknown			
yes			
Chemotherapy	0.001 *		
no/unknown		Reference	
yes		0.296 (0.138–0.633)	0.002 *

SEER, the Surveillance, Epidemiology, and End Results program; yrs, years old; CI, confidence interval; mm, millimeters. Log rank *p* < 0.05 indicates statistical significance. * represents *p* < 0.05.

**Table 4 jcm-12-02521-t004:** Point assignment and prognostic scores in the initial nomogram.

Prognostic Factors	Score
Liver metastasis	
no	0
yes	10
Lung metastasis	
no	0
yes	4.5
Chemotherapy	
no/unknown	4.5
yes	0

**Table 5 jcm-12-02521-t005:** Predictive factors for overall survival of patients with PSOS in Changzheng Hospital.

Prognostic	Univariate Analysis	Multivariate Analysis	
Factors	*p* Value	Hazard Ratio (95% CI)	*p* Value
Laterality	0.396		
left			
right			
not paired			
Grade	0.002 *		
I		Reference	
II		0.774 (0.333–1.801)	0.552
III and IV		1.383 (0.560–3.416)	0.482
T	0.077		
T1			
T2			
T3			
T4			
Liver metastasis	1		
no			
yes			
Lung metastasis	<0.0001 *		
no		Reference	
yes		3.673 (1.525–8.846)	0.004 *
Stage	<0.0001 *		
localized		Reference	
regional		1.145 (0.616–2.129)	0.668
distant		2.110 (1.014–4.387)	0.046 *
Involving segments	<0.0001 *		
single		Reference	
multiple		3.742 (1.802–7.770)	<0.0001 *
ALP	<0.0001 *		
normal		Reference	
abnormal		1.799 (1.043–3.105)	0.035 *
LDH	<0.0001 *		
normal		Reference	
abnormal		1.442 (0.766–2.714)	0.256
Smoking history	0.389		
no			
yes			
Loss of weight	0.002 *		
<5 kg		Reference	
≥5 kg		1.405 (0.823–2.402)	0.213
Frankel	0.815		
A–C			
D–E			
ECOG	<0.0001 *		
0–2		Reference	
3–4		2.573 (1.373–4.822)	0.003 *
Radiotherapy	0.952		
no			
yes			
Surgery	0.004 *		
no		Reference	
yes		0.396 (0.145–1.080)	0.07
En bloc resection	<0.0001 *		
no		Reference	
yes		0.400 (0.177–0.904)	0.028 *

PSOS, primary pelvis and spine osteosarcoma; ALP, alkaline phosphatase; LDH, lactate dehydrogenase; kg, kilograms; ECOG, Eastern Cooperative Oncology Group. Log rank *p* < 0.05 indicates statistical significance. * represents *p* < 0.05.

**Table 6 jcm-12-02521-t006:** Point assignment and prognostic scores in Changzheng nomogram.

Prognostic Factors	Score
Lung metastasis	
no	0
yes	10
Involving segments	
single	0
multiple	7.5
ALP	
normal	0
abnormal	3.5
ECOG	
0–2	0
3–4	6
En bloc resection	
yes	0
no	7

ALP, alkaline phosphatase; ECOG, Eastern Cooperative Oncology Group.

## Data Availability

Publicly available datasets were analyzed in this study. These data can be found here: http://seer.cancer.gov/, accessed on 11 May 2022.

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
