# Peer review of "Development and Validation of a Prognostic Model for Overall Survival in Patients with Primary Pelvis and Spine Osteosarcoma: A Population-Based Study and External Validation"

_jcm, 2023, doi:10.3390/jcm12072521_

Round 1

Reviewer 1 Report

 Dear Editor,

the authors in this paper  developed a predictive model that can effectively predict OS of PSOS patients, with satisfactory performance and clinical practicability. 

Howver some points need to be improved

Since the authors add  as marker  en block resection please better specify which is  its  possible role to worse the prognosis

Please add as marker of poor prognosi of osteosarcoma  also molecular marker as   doi: 10.1186/1471-2407-11-472.

Some comments on

There are errors in Table nnumbers and the  quality of figures 4 and 7 is poor need to be improved

several typing errors and editing english

Author Response

1.Since the authors add  as marker  en bloc resection please better specify which is  its  possible role to worse the prognosis

Response:

We think this is an excellent suggestion. We have added the possible role of en bloc resection to worsen the prognosis into the Discussion part in the revised manuscript (Page 19, Line 41-51).

However, the morbidity of en bloc resection in the spine should always be considered in the decision-making process. The complications of en bloc resection may affect the patient’s quality of life and even worsen the prognosis. Boriani et al. performed a retrospective study of 134 patients who had undergone en bloc resection in the spine. Forty-seven on the 134 patients (34.3%) suffered a total of 70 complications. Three patients (2.2%) died from complications. Demura et al. performed a retrospective study of 307 patients who underwent total en bloc spondylectomy in a single centre. Major and minor operative complications were observed in 122 (39.7%) and 84 (27.4%) patients respectively. These complications included bleeding more than 2,000 ml in 60 (19.5%) patients, cerebrospinal fluid leakage in 45 (14.7%), respiratory in 52 (16.9%), cardiovascular in 11 (3.6%). Any of these complications may be fatal if not managed effectively.

  1. Please add as marker of poor prognosis of osteosarcoma also molecular marker as doi: 10.1186/1471-2407-11-472.

Response:

We sincerely appreciate this valuable comment. We have added markers associated with poor prognosis in patients suffering osteosarcoma, such as YY1 into the Discussion part (Page 19, Line 11-21).

With the rapid development of technology, more and more studies have reported novel markers for poor prognosis of osteosarcoma. Nigris et al. established that overexpression of the polycomb transcription factor Yin Yang 1 (YY1) in primary site of osteosarcoma is associated with poor prognosis, which may be a novel marker for patients with osteosarcoma. Zhu et al. identified circular RNAs (circRNAs), hsa_circ_0081001 and circPVT1, were significantly up-regulated in osteosarcoma and associated with poor overall survival. We believe that novel markers will be used as part of routine clinical practice in the future, complementing the prognostic model for PSOS patients.

  1. There are errors in Table numbers and the quality of figures 4 and 7 is poor need to be improved

Response

We have rearranged the table numbers and improved the quality of figure 4 and 7 in the revised manuscript.

  1. several typing errors and editing English

Response

Thanks for your careful checks. We are sorry for our carelessness. Our revised manuscript has undergone English language editing by MDPI. The text has been checked for correct use of grammar and common technical terms, and edited to a level suitable for reporting research in a scholarly journal.

Reviewer 2 Report

This paper is aimed to develop and validate a Prognostic Model for Overall Survival in Patients with Primary Pelvis and Spine Osteosarcoma based on two population datasets. Overall is a well-written and easy to read, I only have some minor comments:

- In the Introduction, nomogram is briefly mentioned, please give the rationale to use in this paper.

- Results are clear and detailed, but tables's order need to be re-arranged (table 4 follows table 1), and redundancy in Text/Tables1&4 must be checked.

- Discussion includes limitations and a brief analysis on the differences of the two datasets as a reason for the low-performing initial nomogram, but it is not clear how the differences in diagnosis and treatment (from two different countries) could be explaining the differences of the two datasets.

Author Response

1.In the Introduction, nomogram is briefly mentioned, please give the rationale to use in this paper.

Response:

We have complemented the rationale of nomogram into the Introduction part in the revised manuscript (Page 2, Line 22-29).

A nomogram is a graphical depiction of a prediction model that can be applied to assess the overall probability of a specific outcome for any individual. The graph integrates all kinds of important factors and provides a personalized estimate of the probability of events based on regression models. Moreover, the nomogram, as a graphical representation, is convenient to utilize in practice and has been confirmed to be a reliable instrument in the clinical prediction of liver cancer, lung cancer, and breast cancer.

  1. Results are clear and detailed, but tables's order need to be re-arranged (table 4 follows table 1), and redundancy in Text/Tables1&4 must be checked.

Response:

We have rearranged the table number in the revised manuscript.

  1. Discussion includes limitations and a brief analysis on the differences of the two datasets as a reason for the low-performing initial nomogram, but it is not clear how the differences in diagnosis and treatment (from two different countries) could be explaining the differences of the two datasets

Response

Thanks for the reviewer’s question. It is a key point and we would like to explain it in the reply.

In fact, all prognostic models are constructed based on the training datasets. Data quality of the training dataset is crucial to the quality of final model. As the old saying goes, garbage in, garbage out. The most commonly used definition of data quality is “fitness for use for a specific purpose”. The aim of our study is to develop a model for prognosis estimation in PSOS patients. PSOS is a rare malignancy and hard to treat. Treatment method is an important factor affecting the prognosis of PSOS patients, which is also an integral part of our study. Although the SEER database contains multi-institutional data, treatment of rare malignant tumor is its weak point. Spine Tumor Center of Changzheng hospital is one of the largest facilities in China dedicated to the treatment of spinal tumors, which ensures that its data accurately reflects the current diagnosis and treatment of PSOS. Therefore, Changzheng dataset is perfectly suitable to our purpose.